# The Diagnostic and Predictive Value of ^18^F-Fluorodeoxyglucose Positron Emission Tomography/Computed Tomography in Laryngeal Squamous Cell Carcinoma

**DOI:** 10.3390/cancers15225461

**Published:** 2023-11-17

**Authors:** Akram Al-Ibraheem, Ahmed Saad Abdlkadir, Qaid Ahmed Shagera, Omar Saraireh, Dhuha Al-Adhami, Rakan Al-Rashdan, Farah Anwar, Serin Moghrabi, Issa Mohamad, Kristoff Muylle, Enrique Estrada, Diana Paez, Asem Mansour, Egesta Lopci

**Affiliations:** 1Department of Nuclear Medicine and PET/CT, King Hussein Cancer Center (KHCC), Amman 11941, Jordan; ahmedshukri92@hotmail.com (A.S.A.); duha92ali@hotmail.com (D.A.-A.); f.anwar@warith-ici.net (F.A.);; 2School of Medicine, University of Jordan, Amman 11942, Jordan; 3Nuclear Medicine Department, Institut Jules Bordet, Erasme Hospital, Hôpital Universitaire de Bruxelles, Université Libre de Bruxelles, 1070 Brussels, Belgium; dr.qaid22@gmail.com; 4Department of Surgical Oncology, King Hussein Cancer Center (KHCC), Amman 11941, Jordan; oa.11038@khcc.jo; 5Department of Nuclear Medicine, Warith International Cancer Institute, Karbala 56001, Iraq; 6Department of Radiation Oncology, King Hussein Cancer Center (KHCC), Amman 11941, Jordan; imohamad@khcc.jo; 7Department of Nuclear Medicine, AZ Delta, 8800 Roeselare, Belgium; kristoff.muylle@ulb.be; 8Nuclear Medicine and Diagnostic Imaging Section, Department of Nuclear Sciences and Applications, International Atomic Energy Agency (IAEA), 6CM8+ Vienna, Austria; e.estrada-lobato@iaea.org (E.E.); d.paez@iaea.org (D.P.); 9Department of Diagnostic Radiology, King Hussein Cancer Center (KHCC), Amman 11941, Jordan; 10Nuclear Medicine Unit, IRCCS, Humanitas Clinical and Research Hospital, Via Manzoni56, 20089 Milan, Italy; egesta.lopci@gmail.com

**Keywords:** FDG PET/CT, laryngeal neoplasms, squamous cell carcinoma, lymph nodes, MTV, TLG, lymphatic metastasis

## Abstract

**Simple Summary:**

This retrospective study compares the diagnostic efficacy of ^18^F-fluorodeoxyglucose positron emission tomography/computed tomography (^18^F-FDG PET/CT) in detecting nodal disease with that of neck magnetic resonance imaging (MRI). It then measures the rate of change in therapy intent when relying on ^18^F-FDG PET/CT nodal staging results. In a group of 66 patients, our findings emphasize the importance of metabolic tumor volume and nodal size in distinguishing between benign and metastatic lymph nodes. These identified parameters present a promising avenue for reliably predicting nodal disease status, thereby offering robust imaging-based support for future research endeavors in this domain.

**Abstract:**

This retrospective study examines the diagnostic accuracy of ^18^F-fluorodeoxyglucose positron emission tomography/computed tomography (^18^F-FDG PET/CT) and neck magnetic resonance imaging (MRI) in detecting nodal metastasis for patients with laryngeal squamous cell carcinoma (LSCC) and assesses the predictive values of metabolic and structural features derived from ^18^F-FDG PET/CT. By involving 66 patients from 2014 to 2021, the sensitivity and specificity of both modalities were calculated. ^18^F-FDG PET/CT outperforms neck MRI for nodal disease detection, with 89% sensitivity, 65% specificity, and 77% accuracy for nodal metastasis (*p* = 0.03). On the other hand, neck MRI had 66% sensitivity, 62% specificity, and 64% accuracy. Approximately 11% of patients witnessed a change in their therapy intent when relying on ^18^F-FDG PET/CT nodal staging results. Analyzing the cohort for PET-derived metabolic and morphological parameters, a total of 167 lymph nodes (LN) were visualized. Parameters such as the LN maximum standardized uptake value (SUVmax), metabolic tumor volume (MTV), total lesion glycolysis (TLG), and LN size were computed. Logistic regression and receiver operating characteristic (ROC) analyses were performed. Among the 167 identified cervical LNs, 111 were histopathologically confirmed as positive. ROC analysis revealed the highest area under the curve for LN MTV (0.89; *p* < 0.01), followed by LN size (0.87; *p* < 0.01). Both MTV and LN size independently predicted LN metastasis through multivariate analysis. In addition, LN MTV can reliably predict false-positive LNs in preoperative staging, offering a promising imaging-based approach for further exploration.

## 1. Introduction

Accurate lymph node (LN) evaluation is a crucial step in diagnosing and managing laryngeal squamous cell carcinoma (LSCC) [1]. Conventional imaging modalities such as computed tomography (CT) and magnetic resonance imaging (MRI) techniques are established in the guidelines as standard modalities for evaluating the primary tumor extension [2]. However, their role in accurately evaluating nodal extension is limited because they rely solely on the LN morphologic features and cannot evaluate small metastatic LNs [3]. In contrast, ^18^F-fluorodeoxyglucose positron emission tomography/computed tomography (^18^F-FDG PET/CT) has been shown to have higher specificity, sensitivity, and negative predictive value than conventional imaging in identifying metastatic LNs [4]. The unique advantage of the hybrid modality of providing both metabolic and morphologic information makes it more accurate in identifying metastatic LNs [5]. However, the lack of specificity poses a significant challenge in achieving reliable nodal staging.

Previous studies have employed different approaches to discriminate or identify metastatic LNs in cases of head and neck cancers (HNCs), relying on a predetermined SUV threshold [6,7,8]. Promising results were shared. However, all of these studies did not take into account specific HNC subtypes and were not interested in examining volumetric parameters, limiting their final results. Notably, these limitations can complicate the observed results and introduce inconsistencies.

The primary objective of this retrospective study is to investigate the diagnostic and predictive utility of ^18^F-FDG PET/CT and evaluate PET-derived parameters that can effectively identify false-positive LNs, thereby improving specificity.

## 2. Materials and Methods

### 2.1. Patients

The Institutional Review Board (IRB) of King Hussein Cancer Center (KHCC) approved this study (ID: 22 KHCC 123). Informed consent was waived because of the retrospective nature of the study. The study was conducted following the Good Clinical Practice and the 1964 Declaration of Helsinki and its later amendments.

A total of 66 patients diagnosed with LSCC from December 2014 to January 2021 were retrospectively enrolled. The patient was included if he/she fulfilled the following inclusion criteria: (1) diagnosed with biopsy-proven LSCC; (2) had baseline ^18^F-FDG PET/CT and neck MRI performed within 4 weeks before surgery; and (3) underwent surgical total laryngectomy with bilateral or unilateral neck dissection. Patients were excluded if they had synchronous or metachronous neoplasms or baseline PET scans performed >4 weeks before surgery.

Clinicopathologic factors, including patients’ gender, age at diagnosis, tumor stage (based on American Joint Committee on Cancer, 8th edition (AJCC 8th) [9]), and surgical intervention data, were collected. The data of the primary tumor, including size, location, grade, and laterality, were registered.

### 2.2. Neck MR Imaging and Interpretation

Magnetic resonance imaging (MRI) of the neck was performed using 1.5-Tesla (1.5 T) and 3-Tesla (3 T) units. A head coil with a width of 30 cm was employed to encompass the region spanning from the frontal sinuses to the C5-C6 level. All patients underwent imaging using axial, sagittal, and coronal T1-weighted turbo spin echo (TSE) sequences, as well as axial and coronal T2-weighted m-DIXON sequences. Additionally, post-contrast material m-DIXON sequences were acquired in axial, coronal, and sagittal planes using T1 weighting. The acquisition of axial T1-weighted fast spin echo (FSE) images involved using a repetition time (TR) of 670 ms and an echo time (TE) of 18 ms. The echo train length was set to 5, and two signals were utilized. The field of view (FOV) was 240 mm, and the matrix size was 225 × 240. Each image section had a thickness of 4 mm, with a 0.4 mm gap between sections. Axial T2-weighted m-DIXON images were obtained using a repetition time (TR) of 2500 ms and an echo time (TE) of 80 ms. The echo train length was set to 19, with a field of view (FOV) of 240 mm. The matrix size was 250 × 380, and the acquired sections had a thickness of 4 mm with a 0.4 mm gap between them. Axial post-contrast material m-DIXON images were obtained using a T1-weighted sequence. The acquisition parameters included a repetition time of 500 ms and an echo time of 7 ms. The echo train length was set to 5, with a field of view of 240 mm and a matrix size of 225 × 225. Each image section had a thickness of 4 mm, with a 0.4 mm gap between sections. In the contrast-enhanced series, a bolus injection of gadolinium-based contrast agent was administered intravenously at a rate of 2 mL/s, with a dosage of 0.2 mmol/kg of body weight. The series of neck MRI studies were evaluated by a proficient radiologist. The acquisition of these images occurred within a maximum interval of two weeks or simultaneously with the acquisition of the ^18^F-FDG PET/CT images. The metastatic status of cervical LNs was determined based on the presence of central necrosis or a heterogeneous enhancement pattern. The determination of metastatic nodal disease relies on the application of diameter cutoff values exceeding 1.5 cm in the jugulodigastric and submandibular regions, and 1 cm in all other levels of cervical LNs [10]. The process is deemed significant when the ratio of the maximum longitudinal nodal length to the maximum axial nodal length is less than 2, or when a cluster of three or more LNs nears the threshold for metastasis. The classification of regional LN groups is based on the locations and boundaries of cervical LN groups, resulting in six distinct levels [11].

### 2.3. ^18^F-FDG PET/CT Imaging Protocol

All patients were asked to fast for at least 4–6 h, and their serum blood sugar was below 11.1 mmol/L. PET/CT images were acquired 60 min after injection of 3–5 MBq/kg of ^18^F-FDG [12]. A comprehensive imaging procedure was conducted, wherein a whole-body CT scan was performed from the vertex to the mid-thigh. This scan utilized a free breathing signal. The ^18^F-FDG PET/CT protocol was implemented utilizing a Biograph mCT 64 PET/CT system (Siemens, Erlanger, Germany). PET reconstructions were conducted with and without attenuation correction. CT image acquisition was performed using a Biograph mCT flow CT scanner (64 slices), and PET images were acquired using Flow Motion technology (Erlangen, Germany). The image reconstruction was performed using the order-subset expectation maximization (OSEM) algorithm. Attenuation correction and anatomical localization were achieved through the utilization of low-dose CT without the administration of intravenous contrast. The thickness of each slice was 5 mm. The acquisition process employed a table speed of 1 mm/s, corresponding to a duration of 3 min per bed position.

### 2.4. Image Analysis and PET Parameter Quantification

Two experienced nuclear medicine specialists blind to the final pathology results analyzed the ^18^F-FDG PET/CT images using Syngo.via software (Version VB40) (Siemens, Erlanger, Germany) (access date 13 August). The region of interest (ROI) was manually drawn around LNs using the fixed percentile (SUV > 40%) contouring method as recommended by the EANM guideline [13]. The PET parameters, including SUVmax, MTV, and TLG (MTV × SUVmean), were automatically generated. Using Syngo.via software, the size of LNs can be measured using the software’s measurement tools. The software provides various measurement options, such as diameter, volume, and SUVmax. Each and every visualized LN was detailed in terms of its morphologic and metabolic features, in addition to site and laterality, to be correlated with histopathologic references.

### 2.5. Statistical Analyses

Conventional statistical analyses were performed. Continuous variables are reported as the median and interquartile range (Q1–Q3), and categorical data are reported as frequencies and percentages. Mann–Whitney U tests were performed for continuous variables to assess differences between metastatic and non-metastatic LN groups. Sensitivity, specificity, and overall accuracy were computed for each diagnostic test. The McNemar test was employed to assess statistical differences between both modalities in nodal staging.

The area under the curve (AUC) and Youden index analyses using the receiver operating characteristic (ROC) curve were performed to determine the optimal cutoff values for continuous variables significantly associated with the LN metastases. Univariate logistic regression analysis was performed to assess the association between the PET parameters and LN metastatic status. Before conducting multivariate logistic regression analysis, the Spearman correlation coefficient (Spearman rho) was used to exclude strong relationships between the obtained variables. Spearman rho exceeding 0.8 indicated strong correlations, while values below 0.5 indicated weak correlations, and other values indicated moderate correlations. Factors with strong correlation were excluded from the multivariable analysis to avoid the collinearity effect.

Finally, to assess the predictive value of PET-derived factors for the LN false-positivity rate, a multivariable logistic regression analysis was performed by incorporating the LN false-positivity rate as a dependent variable. Following this, a decision tree was formulated to construct a user-friendly clinical algorithm for assessing LN false positivity. The analysis employed the exhaustive Chi-squared Automatic Interaction Detector (CHAID) estimation procedure with a 10-fold cross-validation approach, and the results were corroborated using the Chi-squared Residual Tree (CRT) method. A *p*-value lower than 0.05 was employed to attain results of statistical significance. All statistical analyses were performed using SPSS version 27 software (IBM Corporation, Armonk, NY, USA).

### 2.6. Reference Standard

The findings from imaging modalities were compared to histopathologic results, which were considered as the reference standard. A skilled surgeon reviewed the surgical report and identified the chosen surgical method for neck dissection. Likewise, an experienced histopathologist examined the histopathology findings for each surgical procedure. The LN site, size, laterality, and anatomical levels were annotated.

For patient-based analysis, a thorough methodology for evaluating cervical LNs in LSCC was carried out. This approach was primarily intended to calculate the accuracy for both ^18^F-FDG PET/CT and neck MRI modalities. For neck MRI and ^18^F-FDG PET/CT, morphological factors were examined to identify essential characteristics like size, shape, location, and any morphological anomalies within the LNs. ^18^F-FDG metabolic activity was assessed to determine SUVmax and detect hypermetabolic LNs. The findings were then compared with biopsy results to determine the presence or absence of metastatic processes. Patients were classified into four groups: true positives, false positives, true negatives, and false negatives. True outcomes occur when the imaging modality aligns with biopsy findings, while false outcomes occur when conflicting results are documented.

For lesion-based analysis, all LN morphologic and metabolic ^18^F-FDG PET parameters were computed for every visualized LN, irrespective of its metabolic activity. Each visualized LN was compared to the histopathology results of LNs of similar size and anatomical location in the same patient. Importantly, we excluded any visualized LNs that lacked available matching histopathology results and were located in unexplored anatomical levels or sides. Subsequently, these LNs were included in ROC analysis, where the histopathologic findings served as the basis for establishing the threshold cutoff values. These identified threshold values were then utilized in both univariate and multivariate logistic regression analyses. Lastly, the predictive capabilities of these variables were evaluated in the context of LN false positivity, with the false-positivity rate of LNs serving as the dependent variable in the multivariate analysis.

## 3. Results

### 3.1. Patients

The study cohort comprised 66 patients (63 males and 3 females) with a median age of 55 years. The median time between baseline PET/CT and surgery was 17 days (4–30 days). All patients underwent total laryngectomy with unilateral or bilateral neck dissection (Table 1).

### 3.2. Patient-Based Analysis: Diagnostic Accuracy

The use of ^18^F-FDG PET/CT yielded superior results compared to neck MRI, demonstrating higher sensitivity and comparable specificity. The sensitivity, specificity, and accuracy for detecting nodal disease were 89%, 65%, and 77%, respectively, for ^18^F-FDG PET/CT, while for neck MRI, these values were 66%, 62%, and 64%. These findings were deemed statistically significant, with a *p*-value of 0.03 (Table 2).

^18^F-FDG PET/CT was able to achieve correct nodal staging in 50 patients, while neck MRI correctly staged 42 patients. A change in therapy intent was witnessed in 11% of the patients based on ^18^F-FDG PET/CT reports providing reliable morphologic and metabolic information in uncertain scenarios.

### 3.3. Lesion-Based Analysis

#### 3.3.1. Metabolic and Morphologic Parameters

Among all 66 enrolled patients, only 33 had histopathology-proven metastatic LNs visualized on ^18^F-FDG PET/CT. About 42% of them (*n* = 14) had N2 disease, followed by N1 and N3 (*n* = 13, and *n* = 6, respectively). A total of 167 LNs were visualized and depicted via ^18^F-FDG PET/CT. For each one of these visualized LNs, a subset of morphologic and metabolic features was collected and analyzed. Morphologic features like LN size (median of 1.2 and interquartile range of 0.7–1.6), LN shape, LN anatomic level, and LN laterality were examined. Additionally, LN SUVmax (median of 2.7 and interquartile range of 1.9–4.6), LN TLG (median of 3.6 and interquartile range of 1.3–10.6), and LN MTV (median of 1.9 and interquartile range of 0.9–3.5) were also analyzed.

#### 3.3.2. PET/CT Parameters’ Association with Metastatic Lymph Nodes

A total of 167 LNs were detected via PET/CT, with 111 (66.5%) being metastatic and 56 (33.5%) being non-metastatic as determined histopathologically. The metabolic and morphologic LN factors, including SUVmax, MTV, TLG, and size, showed significant association with the metastatic status of the LNs (*p* < 0.0001 for all). Metastatic LNs presented with statistically significantly higher SUVmax, MTV, TLG, and size values (Table 3) than non-metastatic LNs (Mann–Whitney *p* < 0.0001 for all).

#### 3.3.3. Optimal Cutoffs of PET/CT Parameters

ROC curve analyses were performed for the metabolic and morphological parameters to determine their optimal cutoff values (Table 4).

The optimal cutoff values for LN SUVmax, LN size, MTV, and TLG were 3.2, 1.1, 1.2, and 2.8, respectively (areas under the curve of 0.77, 0.87, 0.89, and 0.85, respectively). The MTV presented the highest AUC and sensitivity (97%), while the TLG had the highest specificity, exceeding 80% (Figure 1).

#### 3.3.4. Logistic Regression Analysis

Univariate and multivariate logistic regression analyses were carried out for the obtained LN factors. On univariate analysis, all parameters were found to be significantly predictive of LN metastases in LSCC (*p* < 0.0001 for each). Before performing multivariate analysis, LN TLG was excluded because of its strong correlation with LN MTV and LN SUVmax (spearman rho = 0.87 and 0.84, respectively; *p* < 0.001 for each) to avoid the collinearity effect. Notably, all correlations were found to be statistically significant (*p* < 0.001 for each), and all other obtained factors were moderately correlated (Figure 2).

In multivariate analysis, the LN MTV and size were retained as independent significant predictors of metastatic LNs (Table 5).

#### 3.3.5. Lymph Node False-Positivity Rate

To assess the predictive capabilities of the aforementioned PET-derived parameters for the LN false-positivity rate, a multivariable analysis was conducted by utilizing the LN false-positivity rate as a dependent variable. Among all parameters, LN MTC and LN size emerged as independent predictive factors for the LN false-positivity rate (Table 6).

By utilizing the cutoff threshold of 1.2 in our cohort, LN MTV can help reduce the false-positive LN detection from 56 to 10 (Figure 3).

## 4. Discussion

This study demonstrates the efficacy of ^18^F-FDG PET/CT in diagnosing and distinguishing between benign and metastatic LNs in LSCC. This was made feasible by taking into account a range of metabolic and morphological factors. ^18^F-FDG PET/CT provides high sensitivity for detecting LNs in LSCC patients. Despite this, ^18^F-FDG PET/CT may be associated with false-positive LNs, as reflected by its limited specificity. Therefore, an assessment of the parameters associated with LN false positivity is vital. The primary significance of such evidence lies in its capacity to provide nuclear medicine physicians with a comprehensive understanding of the advantages inherent in adopting a collective approach that transcends the sole reliance on SUVmax and size criteria for a given LN. In clinical practice, the inclusion of all PET-derived parameters in our study served to mitigate the occurrence of false-positive LNs. This reduction in false positives can contribute to the enhancement of the accuracy of this modality and strengthens its potential to influence therapeutic decision-making in clinical settings.

The present study utilized multivariate analysis to examine potential indicators of LN metastasis in patients with LSCC. This analysis revealed that LN MTV and LN size exhibit strong predictive capabilities for the occurrence of LN metastasis. Additionally, the use of LN MTV was found to improve FDG PET/CT specificity in our studied sample. To the best of our current understanding, this study represents an initial exploration of the discriminatory capacities of ^18^F-FDG PET/CT in the identification of metastatic LNs in LSCC. Furthermore, this study provides the most comprehensive analysis of LNs conducted thus far, with the objective of assessing the effectiveness of ^18^F-FDG PET/CT in this context.

Accurate preoperative LN assessment is essential for achieving optimal planning in patients with LSCC [14,15]. MRI is frequently employed to assess nodal disease in terms of morphological factors including size, the existence of central necrosis, and/or the presence of unclear nodal margins [16]. In various studies, the sensitivity and specificity of MRI in detecting neck LN metastases in HNC vary, with reported values ranging from approximately 40% to 80% and from 50% to 99%, respectively [17,18,19,20,21,22]. Several studies have been conducted to assess the diagnostic accuracy of ^18^F-FDG PET/CT in the identification of neck LN metastases. A recent meta-analysis revealed that the combined sensitivity and specificity of ^18^F-FDG PET/CT for detecting nodal disease were 91% and 87%, respectively, on a per-patient basis [23]. In a previous retrospective study, it was found that the accuracy of ^18^F-FDG PET/CT in diagnosing nodal disease was very high, surpassing 95% [24]. Our study also found that ^18^F-FDG PET/CT was highly sensitive in detecting nodal disease when analyzing patients. Furthermore, when compared to neck MRI, ^18^F-FDG PET/CT performed better and showed a significant difference in results. These findings support what was previously found in a recent meta-analysis [25]. Additionally, in our study, ^18^F-FDG PET/CT led to a change in treatment plan for 11% of our patients. However, it is important to note that the specificity of ^18^F-FDG PET/CT was limited due to a high rate of false positives [26]. In routine practice, the analysis of ^18^F-FDG PET/CT images of the head and neck region is challenging [27,28]. The intricate anatomical composition of the head and neck region, its proximity to vital structures, and the potential overlap of physiological and pathological radiotracer uptake patterns make it difficult to accurately differentiate benign from metastatic LNs [29]. Moreover, several LN factors, including size and necrosis, that could lead to low metabolic activity contribute to the high incidence of false-negative findings on PET/CT [30]. Consequently, the misinterpretation caused by these limitations results in inappropriate management decisions. Therefore, finding predictive factors that help evaluate the nodal disease extent accurately in LSCC is clinically meaningful and an unmet need.

The findings of our study indicate that the utilization of LN MTV resulted in a sensitivity of 97% and a specificity of 76.8% for the detection of LN metastasis. This implies that the LN MTV parameter yields more precise outcomes in detecting nodal disease, consequently mitigating the occurrence of false-negative diagnoses as a result of its heightened sensitivity. In alternative terms, LN MTV has the potential to serve as a valuable tool for the purpose of excluding LN metastasis. Furthermore, the LN size and LN TLG parameters exhibited a specificity exceeding 80%. This approach may assist in addressing the problem of inaccurate test outcomes caused by the presence of small benign LN lesions exhibiting slightly increased metabolic activity, ruling out false positives. However, it is crucial to note that nuclear medicine physicians cannot depend solely on a single criterion or factor to consistently differentiate between benign and metastatic LNs. Therefore, in order to effectively implement the most advantageous discrimination strategy, it is recommended to adopt a collective approach rather than a selective one. Due to this rationale, the utilization of ^18^F-FDG PET/CT holds significant promise in discerning nodal disease through the integration of various metabolic and morphologic characteristics.

Previous studies have employed different approaches to discriminate or identify metastatic LNs in cases of HNC, relying on a predetermined SUV threshold. For example, Nakagawa and colleagues conducted a study in which they examined 31 metastatic LNs that were pathologically confirmed [6]. These LNs were visualized using PET/CT in a cohort of 11 patients [6]. The researchers made the observation that utilizing an SUVmax cutoff value of 3.5 yielded a sensitivity of 75% and specificity of 94% in the identification of metastasis in enlarged cervical LNs [6]. In a study conducted by Murakami et al., ROC curve analysis was employed to assess the utility of size-based SUVmax cutoff values in 23 patients with HNC [7]. The authors proposed specific LN SUVmax cutoff values based on the LN maximum diameter [7]. The first LN SUVmax cutoff was 1.9 for LNs with a maximum diameter of less than 1 cm [7]. A larger LN SUVmax threshold of 2.5 was offered for LNs with a diameter between 1 and 1.5 cm [7]. Finally, a third cutoff of 3.0 for LNs measuring greater than 1.5 cm was proposed [7]. The aforementioned cutoff values resulted in a sensitivity of 79% and a specificity of 99% for the detection of cervical LN metastasis [7]. Recently, a retrospective ROC analysis of cervical LN metastases in HNC patients revealed a relatively high SUVmax threshold [8]. An LN SUVmax threshold of 5.8 was found to be statistically significant, with an observed sensitivity of 71.4% and specificity of 72.7% [8]. Moreover, the literature has documented various size criteria for evaluating the enlargement of cervical LNs [16]. A study by Curtin et al. investigated patients with metastatic HNC, finding that metastatic LNs with largest axial diameter greater than 1 cm had a sensitivity of 88% and a specificity of 39% [17]. Furthermore, using a cutoff of 1.5 cm, the sensitivity for detecting metastatic LNs was 56%, and the specificity was 84% [17]. In contrast, the Response Evaluation Criteria In Solid Tumors (RECIST) rely on evaluating LNs by measuring their short axis on axial images [31]. According to these criteria, LNs with a measurement of ≥1.5 cm are considered pathologically enlarged [31]. Additionally, smaller cervical LNs measuring between 1 and 1.4 cm in the short axis are considered to be pathologic non-targets [16]. In an attempt to assess the LN diagnostic efficacy of ^18^F-fluorothymidine (FLT) PET/CT in relation to ^18^F-FDG PET/CT for HNC patients, Schaefferkoetter and colleagues performed a comparative study [32]. They observed that the ^18^F-FLT PET/CT modality exhibited superior performance in the detection of greater numbers of LNs [32]. Nevertheless, ^18^F-FDG PET/CT demonstrated greater accuracy in distinguishing between benign and metastatic LNs through SUVmax cutoffs [32]. Pietrzak et al. employed an alternative methodology to evaluate the discriminative effectiveness of ^18^F-FDG PET/CT by analyzing sequential imaging parameters. The study revealed a significant statistical distinction in delayed SUVmax values between physiological and pathological LNs [33]. All the aforementioned studies did not incorporate a thorough examination of all morphologic and metabolic factors. Additionally, they acknowledged limitations arising from small sample sizes, biases in parameter selection, and the presence of tumor heterogeneity. Notably, these limitations can complicate the observed results and introduce inconsistencies.

The present study has some limitations, including its retrospective nature and single-centric experience. Moreover, the assessment of morphological aspects is hindered by the dependence on low-dose CT scanners integrated with PET/CT consoles. Nonetheless, it remains the first and only study to examine the discriminative power of ^18^F-FDG PET/CT specifically for LNs in LSCC patients.

## 5. Conclusions

Our findings indicate that LN parameters (specifically LN MTV and LN size) can be useful predictors for discriminating benign from metastatic LNs in patients with LSCC. These intriguing results have the potential to stimulate further research, allowing for the formulation of predictive criteria that incorporate a comprehensive approach encompassing all significant factors. Therefore, more research work is needed to further support and advance such an important value.

## Figures and Tables

**Figure 1 cancers-15-05461-f001:**
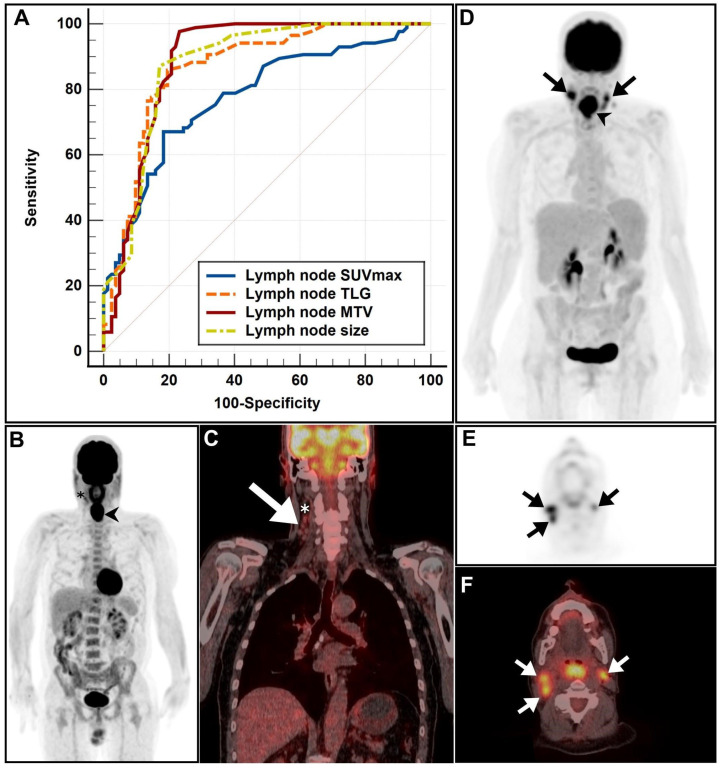
(**A**) The receiver operating characteristic (ROC) analysis of lymph node (LN) factors, including LN maximum standardized uptake value (SUVmax), LN size, LN total lesion glycolysis (TLG), and LN metabolic tumor volume (MTV). (**B**) A maximum-intensity projection (MIP) image in a patient with histopathology-proven N0 laryngeal carcinoma demonstrated evidence of a large hypermetabolic mass confined to the larynx (arrowhead), in addition to a single mildly hypermetabolic right level II cervical LN (asterisk). (**C**) A coronal positron emission tomography/computed tomography (PET/CT) image revealed a cluster of 5 variably sized right cervical LNs (arrow), all of which were subcentimetric apart from the uppermost prominent LN (asterisk), measuring about 1 cm in the largest dimension and appearing mildly hypermetabolic (slightly above the liver SUVmax reference of 2.1). It is noteworthy that all visible LNs in this patient were below ROC cutoffs for size, SUVmax, MTV, and TLG. (**D**–**F**) MIP, axial PET, and axial PET/CT images of a patient with histopathology-proven N3 disease demonstrated evidence of bilateral hypermetabolic cervical lymphadenopathy (arrows), appearing in concordance with a large hypermetabolic laryngeal mass (arrowhead). All observed morphologic and metabolic metrics exceeded the optimal ROC cutoffs obtained from this cohort.

**Figure 2 cancers-15-05461-f002:**
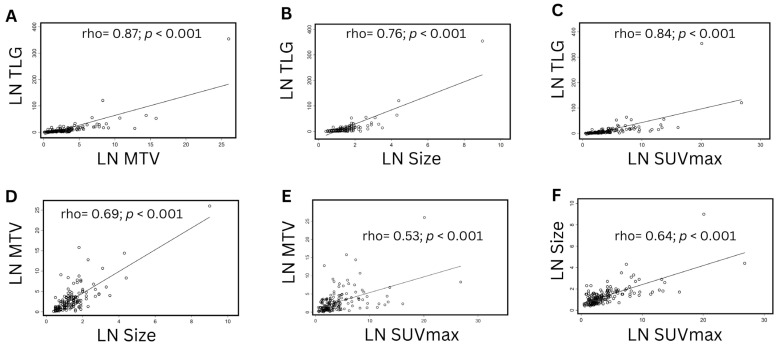
Scatterplots to visualize the correlations between the obtained factors. (**A**) LN TLG vs. LN MTV. (**B**) LN TLG vs. LN size. (**C**) LN TLG vs. LN SUVmax. (**D**) LN MTV vs. LN size. (**E**) LN MTV vs. LN SUVmax. (**F**) LN size vs. LN SUVmax.

**Figure 3 cancers-15-05461-f003:**
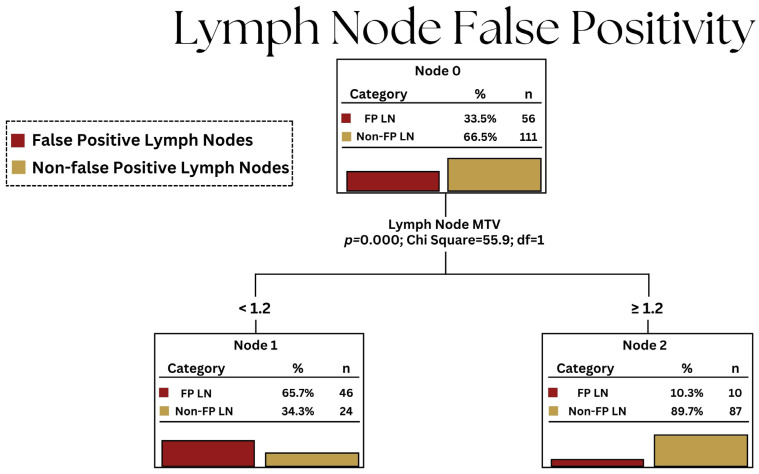
Decision tree illustration of the predictive capability of LN MTV for LN false positivity. The analysis was performed by using the exhaustive Chi-squared Automatic Interaction Detector (CHAID) estimation procedure with a 10-fold cross-validation approach, and the results were corroborated using the Chi-squared Residual Tree (CRT) method.

**Table 1 cancers-15-05461-t001:** Demographic, histopathological, and clinical characteristics of the study sample.

Demographics	
Age (Years)	
Median (Interquartile Range)	55 (49–63)
Gender (Number, Percentage)	
Male	63 (95.5%)
Female	3 (4.5%)
**Histopathologic Characteristics**	
Tumor Site (Number, Percentage)	
Glottic	29 (44%)
Supraglottic	13 (19.8%)
Glottic with supraglottic extension	10 (15%)
Glottic with subglottic extension	3 (4.5%)
Subglottic	3 (4.5%)
Extensive (all compartments involved)	8 (12.2%)
Tumor Size (cm, median and interquartile range)	4 cm (2.5–5 cm)
N-staging	
N0	33 (50%)
N1	13 (19.6%)
N2a	1 (1.5%)
N2b	7 (10.7%)
N2c	6 (9.1%)
N3	6 (9.1%)
Tumor Staging	
Stage III	5 (7.6%)
Stage IVA	53 (80.3%)
Stage IVB	8 (12.1%)
Tumor Grade (Number, Percentage)	
Well-differentiated (G1)	6 (9.1%)
Moderately differentiated (G2)	40 (60.6%)
Poorly differentiated (G3)	20 (30.3%)
Neck dissection	
Bilateral	47 (71.2%)
Unilateral	19 (28.8%)

**Table 2 cancers-15-05461-t002:** Diagnostic accuracy for ^18^F-FDG PET/CT and neck MRI.

^18^F-FDG PET/CT	Neck MRI
- TP ^1^: 26	- TP: 19
- FN ^2^: 3	- FN: 10
- TN ^3^: 24	- TN: 23
- FP ^4^: 13	- FP: 14
- Sensitivity: 89%	- Sensitivity: 66%
- Specificity: 65%	- Specificity: 62%
- Accuracy: 77%	- Accuracy: 64%
**McNemar Test**	
0.03	

^1^ TP: true positive; ^2^ TN: true negative; ^3^ FP: false positive; ^4^ FN: false negative.

**Table 3 cancers-15-05461-t003:** Important anatomic, metabolic, and morphologic features retrieved from the study sample.

Characteristics	Total		Non-Metastatic n. (%)	Metastatic n. (%)
Anatomic Features for the Lymph Nodes (LNs)				
Total LNs	167		56 (33.5%)	111 (66.5%)
Right-sided	83		26 (31.3%)	57 (68.7%)
Left-sided	84		30 (35.7%)	54 (64.3%)
PET/CT Parameters (median and interquartile range)				
Parameter	Total	Negative	Positive	*p*-Value
SUVmax ^1^	2.7 (1.9–4.6)	1.9 (1.4–2.4)	4.8 (4–7.9)	<0.0001
MTV ^2^	1.9 (0.9–3.5)	0.7 (0.4–1)	3.1 (2–4.5)	<0.0001
TLG ^3^	3.6 (1.3–10.6)	1.2 (0.7–1.7)	8.9 (4.6–17.2)	<0.0001
LN Size	1.2 (0.7–1.6)	0.7 (0.6–0.9)	1.5 (1.3–2)	<0.0001

^1^ SUVmax: maximum standardized uptake value; ^2^ MTV: metabolic tumor volume; ^3^ TLG: total lesion glycolysis.

**Table 4 cancers-15-05461-t004:** Metabolic and morphologic parameter cutoffs calculated using receiver operating characteristic analysis.

LN Parameter	Cutoff	AUC ^1^ (95% CI ^2^)	*p*-Value	Sensitivity	Specificity
SUVmax ^3^	3.2	0.77 (0.71–0.83)	<0.001	67.1%	80%
MTV ^4^	1.2	0.89 (0.83–0.93)	<0.001	97%	76.8%
TLG ^5^	2.8	0.85 (0.81–0.91)	<0.001	85.1%	81.7%
LN Size	1.1	0.87 (0.82–0.92)	<0.001	87.1%	81.9%

^1^ AUC: area under the curve; ^2^ CI: confidence interval; ^3^ SUVmax: maximum standardized uptake value; ^4^ MTV: metabolic tumor volume; ^5^ TLG: total lesion glycolysis.

**Table 5 cancers-15-05461-t005:** Logistic regression analysis for predictors associated with metastatic LNs involved in LSCC.

Univariate Analysis			
LN Parameter	Odds Ratio	95% Confidence Interval	*p*-Value
SUVmax ^1^	9.1	4.5–18.7	<0.0001
MTV ^2^	137.6	30.9–612.6	<0.0001
TLG ^3^	1.2	1.1–1.3	<0.0001
LN Size	32.7	13.8–76.9	<0.0001
**Multivariate Analysis**			
**LN Parameter**	**Odds Ratio**	**95% Confidence Interval**	***p*-Value**
SUVmax	2.3	0.8–7	<0.0001
MTV	57.5	12–280	<0.0001
LN Size	8.6	2.7–26.5	<0.0001

^1^ SUVmax: maximum standardized uptake value; ^2^ MTV: metabolic tumor volume; ^3^ TLG: total lesion glycolysis.

**Table 6 cancers-15-05461-t006:** Binary logistic regression model predicting the LN false-positivity rate using PET-derived parameters.

LN Parameter	Odds Ratio	95% Confidence Interval	*p*-Value
LN MTV	1.2	0.4–4	<0.0001
LN Size	0.2	0.1–0.5	<0.0001

## Data Availability

The data presented in this study are available on request from the corresponding author. The data are not publicly available due to privacy.

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
