# Peer review of "The Diagnostic and Predictive Value of 18F-Fluorodeoxyglucose Positron Emission Tomography/Computed Tomography in Laryngeal Squamous Cell Carcinoma"

_cancers, 2023, doi:10.3390/cancers15225461_

Round 1

Reviewer 1 Report (Previous Reviewer 1)

Comments and Suggestions for Authors

A good detailed discussion of MRI and PET criteria, but clinicians in close cases do not rely on absolute numbers but rather short axis size, physiological activity, necrosis, etc. Perhaps rather than including the obvious nodes, concentrating on the specific uncertain ones would be of greater value.

as retrospective and no impact on clinical decisions, perhaps a clearer discussion of utility would be in order 

importantly, how did you identify in pathology specific nodes, especially the non-obvious ones? As above, this is critical to understanding the statistics, especially in a small series.

Finally, as you accurately review, clinicians know where MRI and where PET can be a close call. How does this paper impact that? Saying PET is a bit more helpful than MRI isn’t new

Author Response

Dear Reviewer 1
Thank you  for this informative review shared along with respectful review points and vital questions that necessitate further improvement in many aspects of our article.
Kindly note that we have chosen to subdivide each question with number labeling to facilitate the review process and answer each question accordingly.

Below are the answers to your respectful review points.

  1. A good detailed discussion of MRI and PET criteria, but clinicians in close cases do not rely on absolute numbers but rather short axis size, physiological activity, necrosis, etc. Perhaps rather than including the obvious nodes, concentrating on the specific uncertain ones would be of greater value.
    - We thank the reviewer for making such interesting comments to our manuscript. Indeed, we agree that clinicians do not rely on absolute numbers when dealing with LN metastases; however, the criteria we used were based on previously recommended interpretation criteria. Moreover, adding new sub analysis for a specific list of lymph nodes could introduce selection bias, further reduce the already limited sample size, and, most significantly, contradict the current study's design. We think that the present approach and numbers would bring another tool to nuclear medicine specialists and radiologist, who are interpreting PET/CT images and hopefully increase their confidence in discriminating some equivocal and uncertain nodes. We believe that the present version of our work will serve as a catalyst for future research and help shed light on the important aspect you have raised.

  2. As retrospective and no impact on clinical decisions, perhaps a clearer discussion of utility would be in order
    - We have added a new paragraph to address your respectful concern. Kindly track changes in lines 305-312.

  3. Importantly, how did you identify in pathology specific nodes, especially the non-obvious ones? As above, this is critical to understanding the statistics, especially in a small series.
    - A new section titled “Reference Standard” have been added to address your respectful comment. Kindly track changes in lines 176-202 (highlighted in gray). We have also included an explanation to your inquiry about non-obvious lymph nodes (can be found in lines 193-197; highlighted in turquoise).

  4. Finally, as you accurately review, clinicians know where MRI and where PET can be a close call. How does this paper impact that? Saying PET is a bit more helpful than MRI isn’t new
    - Thank you for this important comment. Actually, in contrast to prior research, our study excluded tumor heterogeneity as a limiting factor by exclusively enrolling LSCC patients. Our results concerning PET/CT metabolic predictors would increase the confidence in PET/CT report and reassure clinicians. As for clinical impact, calculating accuracy for both modalities have helped us find the discordant results and measure the rate of change in therapy intent, relying on FDG PET/CT superiority in instances where MRI understated some patients.

Reviewer 2 Report (Previous Reviewer 2)

Comments and Suggestions for Authors

The authors addressed all comments I’ve made except:

1      1. How was the matching of information from PET-CT scans with histopathological information performed? The description of the methodology is missing! (this is crucial information as it is a potential source of error: namely, the authors did not simply determine the true positivity of one side of the neck or the other, but of the individual lymph nodes!)

In their response added in lines 135-137, the authors only describe FDG PET-CT protocol but not how they matched the PET-CT information with histopathological information.

In this respect, the sentence (lines 150-152; by the way, this sentence is repeated twice) “Each and every visualized LN were detailed in terms of its morphologic, and metabolic features in addition to site and laterality to be correlated with histopathologic reference.” should be supplemented with the description of “histopathological reference” (how the neck specimen was processed by surgeon, how it was processed by the pathologist and how he reported the findings…).

2    2. What was the degree of correlation between LN MTV and LN size?

The authors’ answer (moderate correlation) is not adequate. The readers must be informed abut the details because one would intuitively expect that there is a relationship between metabolic volume and physical volume of LNs (for a similar reason, the authors removed TLG from the multivariate analysis).

Author Response

Dear Reviewer 2
Thank you very your insightful comments. Kindly note that we have chosen to subdivide each question with number labeling to facilitate the review process and answer each question accordingly.

Below are the answers to your respectful review points.

  1. How was the matching of information from PET-CT scans with histopathological information performed? The description of the methodology is missing! (this is crucial information as it is a potential source of error: namely, the authors did not simply determine the true positivity of one side of the neck or the other, but of the individual lymph nodes!). In their response added in lines 135-137, the authors only describe FDG PET-CT protocol but not how they matched the PET-CT information with histopathological information. It should be supplemented with the description of “histopathological reference” (how the neck specimen was processed by surgeon, how it was processed by the pathologist and how he reported the findings…).
    - We have performed two modes of analysis. At first, a patient-based analysis was conducted to determine diagnostic accuracy for both modalities per patient. Afterwards, lesion-based analysis was carried out to determine the predictive capability of PET-derived parameters. The methodology for both modes was included in the revised version of the manuscript. Kindly track the change in lines 176–202.

  2. In this respect, the sentence (lines 150-152; by the way, this sentence is repeated twice) “Each and every visualized LN were detailed in terms of its morphologic, and metabolic features in addition to site and laterality to be correlated with histopathologic reference.”
    - The duplicated sentence has been removed in response to your respectful comment.

  3. What was the degree of correlation between LN MTV and LN size? The authors’ answer (moderate correlation) is not adequate. The readers must be informed about the details because one would intuitively expect that there is a relationship between metabolic volume and physical volume of LNs (for a similar reason, the authors removed TLG from the multivariate analysis).
    - Thank you very much for pointing this out. We have updated the “Logistic Regression” section to include results for Spearman rho for each factor shared in the new figure (Figure 2). Kindly track changes in lines 272–279 (highlighted in gray). Also, we’ve updated the statistical analysis section to clarify the methodology for correlation testing (which can be tracked in lines 161–166; highlighted in gray).

Round 2

Reviewer 1 Report (Previous Reviewer 1)

Comments and Suggestions for Authors

I appreciate your thoughtful efforts to improve the paper.However it is not possible to change the concerblns raised. This is a small retrospective study in which the nodes in question are not prospectively correlated by pathologist, surgeon, and radiologist. Hence bias is introduced that impact statistics.

 The other concerns about clinicians is less of an issue, but true; and some aspects of the revision hence also overstate importance of the distinctions you are discussing.

a pretty elective study would be more powerful.

This manuscript is a resubmission of an earlier submission. The following is a list of the peer review reports and author responses from that submission.

Round 1

Reviewer 1 Report

Comments and Suggestions for Authors

This is a review of 33 N+ pts with laryngeal SCC that states that which is already known. Size and volume (also seen on MRI/CT), as well as SUV and glycolysis correlate with N+. 

Might be of more interest to discuss the set of PET (+) but histology (-) nodes to see if there is something predictive. 

There are also sections such as the INtroduction that could be much shorter: this isn't a discussion of management of laryngeal SCC but only the value of PET-CT. It might be helpful to discuss what new information you think your report has that would warrant publication.

Author Response

Dear Reviewer 1,

We would like to thank you in advance for this informative review shared along with respectful review points in an effort to enhance the manuscript quality.

  • This is a review of 33 N+ pts with laryngeal SCC that states that which is already known. Size and volume (also seen on MRI/CT), as well as SUV and glycolysis correlate with N+.  Might be of more interest to discuss the set of PET (+) but histology (-) nodes to see if there is something predictive. 
    - We appreciate your insightful suggestion to investigate predictive factors in PET(+) but histology(-) nodes in our laryngeal SCC cohort. Investigating potential predictive markers in this subgroup is indeed a valuable endeavor. However, when conducting lesion-based analyses to determine predictive value, it is imperative that the statistical methodology is structured in a manner that minimizes the risk of selection bias. Traditionally, all visualized lymph nodes are evaluated to represent such an idea. Through this approach, we can evaluate the predictive power from multiple angles by including all lymph nodes, ensuring a more robust and unbiased analysis.

  • There are also sections such as the Introduction that could be much shorter: this isn't a discussion of management of laryngeal SCC but only the value of PET-CT. It might be helpful to discuss what new information you think your report has that would warrant publication.
    - The introduction have been modified in response to your respectful review. Previous sentences in arguing the management was removed. In addition, we have applied more focus on literature-based evidence of predictive value of PET parameters. Kindly track changes in lines 78-83 (Highlighted in yellow).
    - As for conclusive remarks, we have updated the conclusion section to include a suggestive guidance for future studies to pursue. Kindly track changes in lines 293-294 (highlighted in yellow).

Reviewer 2 Report

Comments and Suggestions for Authors

In a series of 66 patients with laryngeal SCC who underwent preoperative PET-CT, the authors determined the reliability of several PET-CT parameters (SUVmax, MTV, LN_size, TLG) in predicting the malignancy/benignancy of lymph nodes (LN) in the neck. They found that LN MTV and LN size were independent predictors of differentiation between benign and malignant LNs in this cancer. The article is quite well written, and the methodology, results, and conclusions are appropriate.

P3, L60-62: Diagnosis of metastatic lymph nodes with CT /MR is not based on lymph node size alone!

P5, L154-155: 33 patients had pN+ disease: how can then “The majority of which (n = 14) have N2 disease, followed by N1 and N3 (n = 14, and n = 6, respectively).”?

P5, L162 (3.3. PET /CT Parameters Association with Metastatic Lymph Nodes): How was the matching of information from PET-CT scans with histopathological information performed? The description of the methodology is missing! (this is crucial information as it is a potential source of error: namely, the authors did not simply determine the true positivity of one side of the neck or the other, but of the individual lymph nodes!)

What was the degree of correlation between LN MTV and LN size?

P8, L225: The specificity of LN TLG is less than 80% (and not “exceeding 80%”).

References: Use the prescribed writing style for references!

Author Response

  • In a series of 66 patients with laryngeal SCC who underwent preoperative PET-CT, the authors determined the reliability of several PET-CT parameters (SUVmax, MTV, LN_size, TLG) in predicting the malignancy/benignancy of lymph nodes (LN) in the neck. They found that LN MTV and LN size were independent predictors of differentiation between benign and malignant LNs in this cancer. The article is quite well written, and the methodology, results, and conclusions are appropriate.
  • Thank you for your insightful comments. It is very motivating to know that our work is appreciated and will hopefully have a positive impact on such an important topic.

  • P3, L60-62: Diagnosis of metastatic lymph nodes with CT /MR is not based on lymph node size alone!
  • This has been corrected in the updated version of the manuscript in response to your respectful review. Kindly track changes in lines 61-62 (highlighted in yellow).

  • P5, L154-155: 33 patients had pN+ disease: how can then “The majority of which (n = 14) have N2 disease, followed by N1 and N3 (n = 14, and n = 6, respectively).”?
  • This has been corrected in the updated version of the manuscript in response to your respectful review. Kindly track changes in line 162-163 (highlighted in yellow).

  • P5, L162 (3.3. PET /CT Parameters Association with Metastatic Lymph Nodes): How was the matching of information from PET-CT scans with histopathological information performed? The description of the methodology is missing! (this is crucial information as it is a potential source of error: namely, the authors did not simply determine the true positivity of one side of the neck or the other, but of the individual lymph nodes!)
  • Thanks to your respectful comment. We have added this important notion in lines 135-137 (highlighted in yellow).

  • What was the degree of correlation between LN MTV and LN size?
  • It revealed a moderate correlation.

  • P8, L225: The specificity of LN TLG is less than 80% (and not “exceeding 80%”).
  • The specificity of TLG was 81.7%. We have corrected duplicated number in Table 3 (typo error). Kindly track changes in Table 3, Fourth row, last column (highlighted in yellow).

  • References: Use the prescribed writing style for references!
    - We usually use the premade reference style of MDPI built-in Endnote software. Upon double checking, we did not find any major style errors. We will be thankful and are willing to make any modifications in style/references once determined. Thank you very much for notifying us.

Round 2

Reviewer 1 Report

Comments and Suggestions for Authors

I appreciate your revision, but my concerns and comments remain. Also clinicians do not rely on a specific SUV value but look at size, morphology, etc

Author Response

Dear Reviewer1,

We would like to thank you for providing time and effort to review our work in an effort to improve the manuscript quality. We have updated the manuscript according to your previous recommendations. We will be very grateful if you would reconsider and recommend this article in its current form.

Below are point-to-point answers to previous concerns.

  1. This is a review of 33 N+ pts with laryngeal SCC that states that which is already known. Size and volume (also seen on MRI/CT), as well as SUV and glycolysis correlate with N+. Might be of more interest to discuss the set of PET (+) but histology (-) nodes to see if there is something predictive.
    - Analysis for lymph node false positivity rate was conducted in response to your respectful review. Kindly track changes in lines 255-267.
  2. There are also sections such as the Introduction that could be much shorter: this isn't a discussion of management of laryngeal SCC but only the value of PET-CT. It might be helpful to discuss what new information you think your report has that would warrant publication.
    - The introduction section was trimmed down to present only relevant information as required.

Reviewer 2 Report

Comments and Suggestions for Authors

The authors took into account all comments and corrected the text accordingly. I have no more new comments.

Author Response

(The authors gave the same response as above.)
